# Less is more: Qualitative insights into physically active learning in secondary math education

María González-Pérez [1,2], Alberto Grao-Cruces [1,2*], Francisco J. Bandera-Campos[1,2], Anna Chalkley [3,4], Daniel Camiletti-Moirón[1,2], David Sánchez-Oliva[1,2,5]

**1** GALENO Research Group, Department of Physical Education, Faculty of Education Sciences, University of Cadiz, Puerto Real, Spain, **2** Instituto de Investigación e Innovación Biomédica de Cádiz (INiBICA), Cadiz, Spain, **3** Faculty of Health Studies, University of Bradford, Bradford, United Kingdom, **4** Centre for Applied Education Research, Wolfson Centre for Applied Health Research, Bradford Teaching Hospitals National Health Service Foundation Trust, Bradford Royal Infirmary, Bradford, United Kingdom, **5** Análisis Comportamental de la Actividad Física y el Deporte (ACAFYDE) Research Group, Department of Didactics of Musical, Plastic and Body Expression, Faculty of Sports Sciences, University of Extremadura, Cáceres, Spain

* alberto.grao@uca.es

## Abstract

### Background

Physically active learning (PAL) has shown promise as a pedagogical approach that combines physical and academic benefits by incorporating physical activity during academic lessons. However, its implementation in secondary schools remains a challenge, highlighting the need to explore teachers' and students' perceptions and experiences.

### Aim

To assess the perceptions and experiences of teachers and students following their participation in a PAL intervention in secondary education.

### Methods

Data were collected through individual interviews with mathematics teachers (n = 5) and six focus groups with students (n = 36) who participated in a 16-week intervention with a weekly outdoor PAL lesson. Data were analyzed using conventional content analysis.

### Results

Teachers reported a general increase in students' motivation. They also noted an improvement in the participation and behavior of some students during PAL lessons. These aspects were also reported by the students themselves. Although teachers and students perceived that the amount of content covered in a PAL lesson was lower compared to traditional lessons, they emphasized that learning was more meaningful due to enhanced retention, facilitated by active engagement and dynamic teaching approaches. Teachers and students also perceived an

**Data availability statement:** All relevant data are within the paper.

**Funding:** This work was supported by The National Plan for Research, Development, and Innovation (R&D&I) from the Spanish Ministry of Science, Innovation and Universities [PID2019-104023RA-I00] (DCM); The Andalusian Plan for R&D&I Regional Ministry of Economy, Knowledge, Enterprise and University of Andalusia [P20_00908] (AGC); and the Junta de Extremadura & Fondos Feder [IB20126] (DSO). MGP and FJBC are supported by a grant from the Spanish Ministry of Education [FPU22/01430 and FPU21/03385 respectively]. The funders had no role in study design, data collection and analysis, decision to publish, or preparation of the manuscript.

**Competing interests:** The authors have declared that no competing interests exist.

improvement in students' cooperation and socialization. Teachers considered PAL to be a feasible pedagogical approach to implement, provided adequate resources were available.

## Conclusions

PAL is seen as a pedagogical approach that brings both social and academic benefits to secondary education. However, to ensure its effectiveness and long-term sustainability, it needs to be integrated into the whole educational community.

## Introduction

Physical inactivity is considered a global epidemic and the leading cause of non-communicable diseases [1]. It is also important to highlight its role in areas other than health, such as its positive effect on various academic indicators such as attention, executive function and academic performance [2,3]. Despite this knowledge, the current levels of physical inactivity and sedentary behavior among children and adolescents are alarming [4,5]. In the Spanish context, this situation is particularly worrying. According to Zapico et al. [6], only 36.2% of Spanish children and adolescents meet the World Health Organization's recommendations for daily physical activity [7]. This trend is also reflected in the school setting, where compliance with the American Heart Association's guidelines, recommending at least 30 minutes of physical activity during the school day [8], is insufficient. In this regard, Grao-Cruces et al. [9] reported that less than a quarter of children and adolescents achieve these recommendations, while between 66% and 81% of their time at school is spend on sedentary behaviors [10].

Although schools can be environments where physical inactivity and sedentary lifestyles prevail, they also represent ideal settings to intervene and promote more active and healthy lifestyles. The structure and resources available in schools provide a unique opportunity to integrate programs that not only have the potential to address the students' physical activity necessities, but also enrich their educational experience in a holistic way [11,12]. Initiatives designed to promote healthy lifestyle habits during the school day are widespread. Particularly, those focused primarily on physical education programs and on modifying the school environment, such as creating or transforming spaces for active recess [13,14]. However, in recent years, there has been a shift towards an innovative approach that integrates physical activity into the core subjects (excluding physical education), known as physically active learning (PAL) [15].

PAL incorporates movement-based activities into the curriculum to transform the traditional sedentary lessons into an active environment. This approach has been shown in primary education to not only contribute to recommended daily physical activity levels [16,17] but also to have positive effects on academic indicators, such as academic performance and time-on-task. Specifically, the scientific literature indicates that PAL can improve academic performance in the subjects in which it is

implemented [15,18]. Additionally, PAL has been associated with increased time-on-task during lessons where it is used [17,19,20].

In the context of secondary education, the results are consistent with those obtained in primary education, where PAL appears to have a positive effect on students' physical activity and academic performance [21–23]. However, research at the secondary level remains limited and often focuses on a narrow range of outcomes, preventing a holistic understanding of their wider educational impact. Therefore, there is still a need to develop intervention studies at this stage of education that explore a wider range of outcomes, such as student engagement, social interaction or long-term retention of learning. Furthermore, to ensure the feasibility and sustainability of such interventions over time, it is crucial to give voice and to understand the experiences and perceptions of the participating teachers and students involved. Understanding their insights can highlight practical challenges and benefits, allowing programs to be refined and adapted for better implementation and uptake [24].

This qualitative perspective has been addressed by several studies in primary education, which focus separately on teachers' perceptions [25,26], students' perceptions [27] and some that combine both perspectives [28,29]. The main findings of these studies indicate an increase in students' engagement and concentration during PAL lessons [25]. They also consider PAL as a method that introduces variation and enjoyment into learning [25,28,29], improves social relationships between peers and promoting quality learning [25,26,29]. However, some teachers have difficulty identifying its cognitive and social benefits when applying it to students with cognitive challenges due to their support needs. Furthermore, although they report that it is another path to learning, they are uncertain about achieving learning objectives [25].

On the other hand, in secondary education, qualitative studies are more limited, and those that do exist independently examine teachers' and students' perceptions. These studies generally report similar findings to those observed in primary education: Teachers perceive the PAL as a pedagogical approach that adds variety to teaching, increases student motivation and engagement, and helps students' attention and concentration [30,31]. Lerum et al. [31] further highlight improvements in student collaboration and participation, while Schmidt et al. [30] point out that teachers sometimes feel overwhelmed by the complexity of integrating physical activity into lessons while meeting learning objectives. Secondary students' experiences of PAL are consistent with teachers' perceptions. Research by Romar et al. [32] shows that students perceive the integration of PAL as an innovative approach that disrupts their sedentary learning. In addition, students reported feeling more alert, focused and able to sustain their attention during PAL lessons.

Despite the emerging research on PAL in secondary education, an important gap in understanding its impact and implementation remains. Furthermore, no study to date has included both students and teacher perspectives together in secondary education. This is important because classrooms are social environments where teaching and learning are shaped by both what teachers *intend* and what students *experience*. To address this gap in literature, it is essential to systematically analyze and compare teachers' and students' perceptions of their experiences of PAL in secondary education. This approach will identify the practical challenges, potential benefits and educational implications associated with its implementation, thus contributing to a more comprehensive understanding of its impact in this educational context. Therefore, the aim of the present study was to assess the perceptions and experiences of teachers and students following their participation in a PAL intervention in secondary education.

## Methods

### Design

This study is part of the ACTIVE CLASS randomized controlled trial (registration number: NCT05891054). The ACTIVE CLASS study was conducted simultaneously in the Spanish provinces of Cadiz and Caceres and included a total of two experimental groups (Active Break intervention group and PAL intervention group) and one control group. More

information regarding the study protocol is available elsewhere [33]. The present study included only the PAL intervention group, which completed a 16-week intervention based on including PAL lessons in the subject of mathematics.

This study protocol has been reviewed and approved by the Bioethics Committees of the Andalusian Government (Cadiz, Spain), and the Bioethics and Biosafety Committees of the University of Extremadura (Caceres, Spain).

### Setting and participants

This study was carried out in the Spanish context, where secondary education is compulsory from 12 to 16 years old and last a total of four academic years. Of the six public secondary schools included in the ACTIVE CLASS study, a randomization process was carried out and two schools were selected, which were assigned to the PAL intervention group and included in the present study. These schools were both located in urban areas and, as part of their existing infrastructure, they had suitable outdoor spaces that could be used for PAL lessons. These included outdoor sports grounds, large natural areas and tarmac areas.

Participant recruitment took place from November 2022 to January 2023. The study was explained to the participants before starting, and the volunteers, parents, or tutors provided written informed consent. A total of 104 students (12–14 years old, n = 45 boys) and five mathematics teachers (two male) participated in the PAL intervention. To be included in the analysis, all teachers involved in the intervention were invited to participate in the interviews. On the other hand, from the total number of students, a sub-sample of 36 (six students per participating class [n = 18 boys]) were invited to participate and completed the focus groups for data collection (Table 1). This sub-sample of students was randomly selected within each participating class, with a balanced gender distribution to ensure homogeneity in the composition of the group. Neither students nor teachers had been exposed to previous PAL experiences.

### Intervention

The intervention consisted of including physical activity in mathematics lessons without modifying the content of the subject for 16 weeks. For this purpose, a weekly PAL lesson, each lasting 55 minutes, was performed outside the classroom, assuming a total of 16 PAL lessons, all of which were conducted outdoors. In these lessons, activities such as moving, jumping, throwing, etc. were integrated into the activities of the mathematics subject. To ensure consistency across all PAL lessons, they were always conducted on the same day of the week and at the same time during the 16-week intervention period.

Each PAL lesson began with the teacher introducing the educational content to be covered in the lesson (i.e., equations, geometry, fractions...). The activities developed in the lesson consisted of incorporating the students' movement into the mathematical exercises normally found in the textbook (i.e., proportionality problems, addition and subtraction of decimal numbers...) through games, collaborative tasks or competitive activities. Specific examples of activities that follow our PAL methodology can be found here. This resource was made available to teachers to support them in implementing PAL lessons during the intervention and to provide them with tools to further develop this pedagogical approach in the future, if deemed appropriate.

The mathematics teacher was responsible for sharing the PAL lessons thought the intervention. However, a specialized technician was present at each session to provide practical support to the teachers and to ensure the correct development of the activities. In addition, lessons were co-designed by the mathematics teachers and members of the research team who had expertise in integrating physical activity into educational settings. Thus, prior to the development of each PAL lesson, a meeting was held between the research team and each mathematics teacher. In these meetings, the curricular content that the teacher wanted to teach in the lesson was discussed and the activities to incorporate physical activity into these lessons. Although the researchers aimed to incorporate physical activity to promote student engagement and potential health benefits, the primary focus remained on ensuring the effective delivery of the mathematics content and maintaining alignment with curriculum objectives. This approach sought to balance the dual benefits of promoting learning and increasing physical activity in a structured way.

**Table 1. Participants in the interviews and focus group: sub-sample overview.**

| Researchers | | | | Students | | | |
|---|---|---|---|---|---|---|---|
| Researcher ID | Age | Gender | Credentials | Subject ID | Age | Gender | Class (Focal Group) |
| | | | | Student 1 | 13 | Male | 1 |
| Researcher 1 | 25 | Female | Doctoral Researcher | Student 2 | 12 | Male | 1 |
| Researcher 2 | 25 | Male | Doctoral Researcher | Student 3 | 12 | Male | 1 |
| | | | | Student 4 | 12 | Female | 1 |
| **Teachers** | | | | Student 5 | 12 | Female | |
| Subject ID | Age | Gender | Class | Student 6 | 12 | Female | 1 |
| Teacher 1 | 28 | Female | 1 | Student 7 | 12 | Male | 2 |
| Teacher 2 | 30 | Female | 4 and 5 | Student 8 | 14 | Male | 2 |
| Teacher 3 | 31 | Male | 2 | Student 9 | 12 | Male | 2 |
| Teacher 4 | 35 | Male | 6 | Student 10 | 12 | Female | 2 |
| Teacher 5 | 49 | Female | 3 | Student 11 | 13 | Female | 2 |
| | | | | Student 12 | 12 | Female | 2 |
| | | | | Student 13 | 12 | Male | 3 |
| | | | | Student 14 | 12 | Male | 3 |
| | | | | Student 15 | 13 | Male | 3 |
| | | | | Student 16 | 12 | Female | 3 |
| | | | | Student 17 | 12 | Female | 3 |
| | | | | Student 18 | 12 | Female | 3 |
| | | | | Student 19 | 14 | Male | 4 |
| | | | | Student 20 | 14 | Male | 4 |
| | | | | Student 21 | 13 | Male | 4 |
| | | | | Student 22 | 13 | Female | 4 |
| | | | | Student 23 | 13 | Female | 4 |
| | | | | Student 24 | 13 | Female | 4 |
| | | | | Student 25 | 13 | Male | 5 |
| | | | | Student 26 | 14 | Male | 5 |
| | | | | Student 27 | 13 | Male | 5 |
| | | | | Student 28 | 13 | Female | 5 |
| | | | | Student 29 | 15 | Female | 5 |
| | | | | Student 30 | 13 | Female | 5 |
| | | | | Student 31 | 14 | Male | 6 |
| | | | | Student 32 | 13 | Male | 6 |
| | | | | Student 33 | 13 | Male | 6 |
| | | | | Student 34 | 13 | Female | 6 |
| | | | | Student 35 | 14 | Female | 6 |
| | | | | Student 36 | 14 | Female | 6 |

Note. The class section refers to the specific class group where each teacher taught their lessons. In the case of students, this same number was also used to differentiate the focus groups.

## Data collection

Semi-structured interviews with teachers (n = 5) and student focus groups (n = 6) were used as techniques for qualitative data collection. Both were conducted after the end of the intervention period of the study (June 2023). These methods are supported in educational research for their ability to provide rich, contextual data [34,35].

The research team collaborated in the development of the interview and focus group guides. A first draft of the questions was developed based on previous PAL research and the team's own experience in implementing and evaluating similar interventions. This draft was then revised and refined through multiple rounds of discussion among all authors, including researchers with expertise in qualitative methods, allowing the team to assess the relevance, clarity and completeness of the questions. In both cases, the questions were focused on describing their experiences with PAL.

Some examples of questions included were: *"What do you think about the pedagogical approach of PAL?", "What are the benefits you think PAL has brought you?", "What disadvantages or barriers do you find in using PAL?", "What differences do you find in terms of work and learning content compared to a traditional lesson?", "How do you think PAL is different from a traditional lesson?".*

In addition, participants were encouraged to express their opinions and experiences on themes beyond the initial guide, promoting the emergence of unanticipated but relevant topics.

The interviews and focus groups on each school were conducted by the same researcher to promote trust and familiarity. Both were conducted in the mathematics department to provide a pleasant and quite environment for the participants. In both cases, the interviews lasted between 20–40 minutes (with an average of 30.6 min) and were audio recorded for later transcription.

## Data analysis

The interviews and focus groups were transcribed verbatim into a Microsoft Word document (Microsoft, Redmond, WA). The data were then analyzed by the same researcher who conducted the interviews and focus groups using the conventional content analysis method of Hsieh and Shannon [36], facilitated by Nvivo 14 software for Windows (QSR, Melbourne, Victoria, Australia). To achieve this, the transcripts were read several times after transcription to ensure that the researcher developed a thorough familiarity with the content of the transcript and a comprehensive understanding of the data. Sentences and text segments that encapsulated teachers' and students' experiences and perceptions of PAL were then identified and highlighted. From these identified units of meaning, inductive reasoning was used to create codes, leading to the development of a map of categories and subcategories that were interrelated and aligned with the study's objectives (Fig 1). For example, in response to the question *"What are the advantages you see to this type of pedagogical*

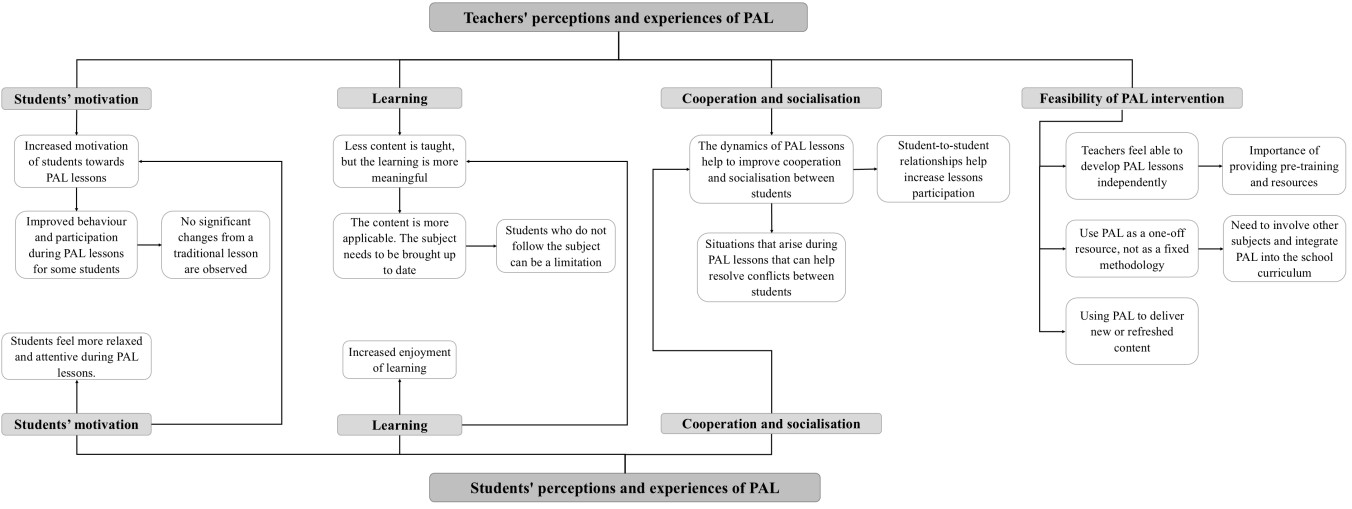

**Fig 1. Flow chart of categories and subcategories analyzed.** Note. PAL = Physically Active Learning.

*approach over a traditional lessons?”,* one teacher stated: *“There are students who are quite nervous, and PAL has been good for them, and there are others who are not good at math and with this increased motivation as competitive activities it has given them a boost (...). Those who hardly do anything in class can even take the role of leaders in the activities”.* This statement was highlighted as a unit of meaning and coded as *“increased motivation”* and *“improved behavior and participation during PAL lessons”.*

These categories were refined and adjusted to accurately represent the data. The process resulted in a structured categorical framework of the data, accompanied by the researchers' interpretation.

To ensure the reliability and credibility of the data, multiple strategies were implemented. First, all interviews and focus groups were conducted by the same researcher in each school, ensuring consistency in the data collection process and reducing variability introduced by different interviewing styles. Second, peer debriefing sessions were held within the research team to discuss, refine, and validate the coding framework and thematic categories. This collaborative process helped minimize researcher bias and enhanced the rigor of the analytical approach.

Although the number of interviews and focus groups was determined in advance based on the study's scope and available resources, data saturation was identified organically during the analysis, when no new themes or categories emerged. Saturation was perceived to have been achieved after the 5th focus group and the 3rd interview, where recurring patterns and consistent themes were observed across both teachers' and students' responses. This repetition of the findings provided confidence that the data adequately captured the range of experiences and perceptions of the participants.

## Results

The results were structured into four main categories. The perceptions collected through interviews and focus groups for each of these categories are detailed below. This approach provides a comprehensive view of the effects and feasibility of PAL in the educational context.

### Motivation of students

This category examines how the implementation of PAL affects learners' motivation and how this can influence their participation and behavior. In this context, motivation refers to students' drive and enthusiasm to actively participate in class. The results showed that the introduction of this pedagogical approach was a methodological novelty for the students, which led to an increase in their motivation. This increase in motivation is evident in the statements of some teachers. Teacher 1 said: *“It is a different way of looking at the subject, so it is a novelty for them. When they see that there are more options, they seem to be more motivated”.* Similarly, Teacher 2 reiterated this observation: *“The first thing is motivation. Participating in games motivates them much more than sitting in the classroom. Anything that is dynamic, that takes them out of the routine and out of the classroom, motivates them more. Motivation allows you to produce learning, which is what we are all about”.* The students themselves were also enthusiastic about the lessons. Students 32, 34 and 36 remarked: *“The physically active lessons were more fun, and we looked forward to the day when we would have to do them”.*

Not only was this increased motivation widespread among students, but some teachers noted that this increased motivation could have a positive impact on the behavior and participation of certain students, as it was an inclusive and attractive form of teaching for them. Teacher 3 said: *“There are students who are quite nervous, and the physically active lessons have been good for them, and there are others who are not good at math and with this increased motivation as competitive activities it has given them a boost (...). Those who hardly do anything in class can even take the role of leaders in the activities”.* This statement was also supported by Teacher 2: *“Those who didn't like to do anything in class also felt much better because they felt less controlled, although I would say they were surprised themselves because some of them ended up participating more than I thought they would”.* Students also reported an improvement in their attention span during these lessons, as evidenced by statement of student 13: *“In the physically active lessons I was not distracted because we were always moving and paying attention”.* And student 24: *“We were less distracted and more focused”.*

However, for other students, while teachers noted an increase in general motivation, this did not necessarily translate into changes in behavior, participation, or engagement. Teachers observed that some students maintained the same level of involvement as in traditional lessons. Teacher 4 explained: *"It is obvious that some were very involved, but others were just passing by, just like in my traditional lessons"*. Teacher 3 confirmed this observation: *"There are students who did not work in class and did not work in the physically active lessons. There is also a group of students who work in class and pay attention, and the physically active lessons were neither good nor bad for them because they kept on doing the same thing"*. These observations suggest that while PAL can increase motivation, its impact on behavioral engagement and participation may vary between students, with some showing no significant change compared to traditional teaching.

### Impact of physically active learning on students' learning

Teachers agree that because of the inclusion of physical activity in PAL lessons, the amount of content and number of activities covered is less than in a traditional classroom lesson, as the physical activities tend to take more time to complete. This perception was clearly expressed by Teacher 3: *"I think the pace of work is slower in the physically active lesson because you lose the pace of the class for the sake of physical activity and motivation"*. Teacher 2 added: *"It's true that you have time to do five exercises in class and two in the physically active lesson, but we get everyone to do it"*. This observation was also generally shared by the students. In this sense, student 21 mentioned: *"I think we worked a bit less in terms of number of activities, but we learned more"*. These perceptions suggest that while the quantity of content covered may be reduced, the quality of the learning experience is enhanced. The slower pace and active engagement allow students to process the content more deeply, promoting understanding and retention.

Despite these initial perceptions, both teachers and students emphasize that the learning achieved is more significant. Teacher 2 stated: *"I saw that they had acquired the content we were working on. In the traditional lesson I often gave them the same thing as in the physically active lesson and I observed that they had learnt the content in question"*. Teacher 3 reinforced this idea by stating: *"I think that the physically active lesson supported student pacing for those who typically struggle with classroom instruction"*. Students also enthusiastically supported this idea, repeatedly mentioning that the PAL lessons helped them to understand the content better, among them, student 2 said: *"I found the physically active lessons fun and at the same time we learnt things that maybe I wouldn't have learnt in the same way in class"*. Student 11 added: *"I didn't understand anything in the traditional classroom lesson, and I understood it better in the physically active lesson"*. And student 16 concluded: *"Playing is a way to learn"*. These reflections suggest that the integration of movement and active engagement in learning may foster a cognitive bonding process, where the physical and social aspects of the activity enhance comprehension and retention of the content.

However, for this learning to take place effectively during a PAL lesson, it is necessary for students to keep up with the subject. Students who do not keep up with the subject can become a hindrance to both the lesson and their peers. Teacher 1 expressed this concern: *"Students who don't keep up with the subject are lost. The same thing happens in the traditional lesson, but because it's an individual task, it doesn't affect the rest of the class"*. Students also highlighted this problem. In this regard, student 28 stated: *"If you are in a group where the others don't do anything, they would take your work, and you would be at a disadvantage"*.

### Cooperation and socialization among students

The move from a traditional classroom layout with students sitting at single or double desks to a PAL environment characterized by group and cooperative activities seems to have had a positive impact on students' cooperation and socialization. Teacher 5 clearly stated: *"They have also benefited in terms of socializing, as it allows them to interact with classmates other than those they normally meet in class"*. Similarly, Teacher 2 added: *In the classroom there was no cooperation between them and in the physically active lessons, with the activities, this relationship with others is set*

*in motion because of the teamwork and the distribution of roles. It allowed them to get to know each other better, which is more difficult in any class unless it is a physical education class"*. Students also confirmed these perceptions. Among them, student 15 noting: *"Sometimes I was with classmates I didn't get along with, so I had to work with them to do the exercise. It improved our relationship"*. Student 6 also expressed: *"Thanks to an exercise I started to trust a colleague and get along better, and before that we hardly spoke to each other"*.

This collaboration has not only improved interpersonal relationships but also seems to have increased participation in lessons, fostering both problem solving and creativity. Teacher 3 observed*: "It was good for some students because there is a group of students who struggle with math and in the physically active lessons they worked as a group and helped each other"*. By working collaboratively, students were not only able to support one another academically but also to approach challenges with creative strategies and shared problem-solving efforts. Teacher 5 expanded on this idea: *"Maybe in the physically active lessons, by doing the activities with peers, by supporting each other in groups, by losing the fear and embarrassment of making mistakes, they tried a bit more to do things"*. Student 3 made a comment in this regard: *"For me, it was good to work in a group and to have my colleagues help me when I didn't know something"*.

In addition, the inherent characteristics of PAL lessons in terms of relationships between students can also help to resolve conflicts that do not arise in a traditional lesson. Teacher 3 highlighted this point: *"In physically active lessons, the relationship with the rest of the classmates is worked on. But it should be well developed, otherwise it can lead to conflicts. However, these conflicts occur more in this type of lesson than in the classroom and this can be positive because it helps to identify problems that are not seen in the classroom and so they can be dealt with and solved"*.

## Feasibility and sustainability of intervention

The last category relates to the participating teachers' perceptions of the feasibility of implementing PAL in secondary schools. The results suggest that after the intervention, teachers feel empowered and confident to autonomously develop PAL lessons in the future. As teacher 1 put it: *"The necessary resources? You gave me more than enough, with all the activities. Also, a lot of the activities we did can be adapted. I think I can do it on my own"*. Teacher 2 also emphasized the importance of the support of the research team and of a previous training period for teacher training*: "I have already gone through the whole process, but it is true that it is complicated for a teacher who has never done it or seen it before. I see the need for more elements to make sure it works, such as the help of a physical education teacher. I've had your help and now I'm daring... but if I hadn't come here, I would have seen the need for this resource"*. This highlights not only the importance of external support, but also the need to develop teachers' confidence and competence. In this sense, confidence, strengthened by practical experience and training, seems to contribute to teachers feeling prepared to implement the PAL autonomously. At the same time, the practical skills acquired may lead to greater competence in adapting PAL to different contexts, reflecting the need for teachers to feel confident and empowered to integrate PAL into their practice.

Another aspect on which teachers agree is that, although they would like to continue using PAL lessons in their subjects, they would do so less frequently. They see PAL as an additional pedagogical resource, useful in specific situations, but not as a fixed methodology. In this sense, teacher 1 said: *"I would do it, like I said, once in a while. What would motivate me to do it is that they ask for it, I know they like it, and it is motivation for them. As long as they are working and revising, I have no problem with it, and it helps them to move around, which is good"*. Teacher 2 added: *"There are some contents that lend themselves more to this kind of methodology and others don't, so I don't see it as a fixed methodology, I see it as a specific resource"*.

Teachers also emphasized the need to extend these interventions to other subjects so that PAL is integrated into the school curriculum. Teacher 2 suggested: *"Personally, I would rotate subjects and aim for three hours of physically active lessons per week across all subjects. Ideally for me it would be daily, one hour per day for each subject. One hour a week seems too little to me, as the benefit is to break up the six hours of sitting"*. Teacher 4 added*: "If more subjects*

*were included, the ideal would be at least once a week, at least in those subjects that have as many hours a week as mathematics".*

Finally, the teachers discussed the most appropriate time to include PAL lessons in their subject. They agreed that PAL lessons tended to be more dynamic when introducing new content or as revision lessons, as they allowed students to better consolidate what they had learned. Teacher 5 said: *"I think it works best for revision because knowing it makes everything go faster".* Teacher 3 added: *"I see these lessons as reinforcement or a very light introduction or a more general idea. I don't see it as a way of teaching new content because it is often difficult for them to understand very simple exercises in themselves. In the classroom you stop and ask questions, but in a physically active lesson they may not have understood, and you may not know... it's more complicated".*

## Discussion

The aim of this study was to explore the perceptions and experiences of teachers and students following their participation in a PAL intervention in secondary education. To date, this study represents the first comprehensive exploration of both teachers' and students' perceptions of PAL in secondary education. The main findings highlight that both teachers and students reported an increase in students' motivation during PAL sessions, leading to increased participation and improved behavior of certain students. Both perspectives agree that although the pace of PAL sessions may be slower, the learning achieved is more substantial. They also emphasize the increased cooperation and socialization between students. Teachers assert that PAL is a viable pedagogical resource for use in educational settings, provided that adequate resources are available, albeit as an occasional teaching tool and not as a fixed approach.

With regard to student motivation, teachers and students reported that PAL lessons were perceived as a novelty resource. This was reflected in increased student motivation, which for some students also correlated positively with improved behavior and participation in activities. These perceptions have been highlighted by previous research in primary education, which has emphasized the importance of positive experiences and the enjoyment of PAL as factors that increase motivation [28,29]. In the context of secondary education, teachers also perceive PAL as a resource that brings variety to the classroom and promotes positive experiences and enjoyment among students, which contributes to greater engagement and motivation for learning [31]. However, it is important to consider whether the motivational benefits of PAL may be partially linked to its novel approach, as the satisfaction of the need for novelty seems to predict intrinsic motivation, as well as students' participation and intention to engage in physical activity [37,38]. As in physical education, this suggests that introducing new and varied proposals, such as PALs, can increase motivation precisely because they satisfy this basic psychological need, and therefore, if overused, its impact on engagement and learning could diminish over time. For this reason, further research is needed to explore the long-term effects of PAL and ensure its sustainability in educational settings. In this regard, Daly-Smith et al. [18] highlight the importance of familiarization with the intervention to reduce novelty effects and maximize its benefits, suggesting that adequate implementation strategies could help maintain its effectiveness over time.

The impact of PAL on learning and retention was a key evaluation point in this study. Although both teachers and students perceive that PAL lessons may cover less content than traditional lessons, the depth and sustainability of the learning achieved appears to be significantly improved. Teachers participating in a PAL intervention have previously highlighted this observation, emphasizing the academic and cognitive benefits of this pedagogical approach for their students [26], Students participating in PAL interventions also agree, reporting that the teacher acts more as a facilitator of learning than in traditional teaching [27]. This could be explained by PAL's consistency with the principles of sticky learning. In this regard, teachers involved in the research by Chalkley et al. [39], highlight that PAL can promote the physical embodiment of learning, thus creating memorable and engaging educational experiences that promote the internalization and retention of knowledge. These findings suggest that, far from compromising learning and academic rigor, PAL can enhance it by fostering a more interactive and engaging learning environment [29].

The above perceptions are supported by quantitative data, which indicate that PAL interventions can produce significant improvements in educational outcomes for primary school children [17]. These results could be transferred to secondary education, where research by González-Pérez et al. [23] shows a significant improvement in time-on-task as academic indicator during and after the implementation of a PAL intervention. However, further research and development of PAL in this context is. In this regard, teachers' perceptions as reflected in the studies by Schmidt et al. [30] and Lerum et al. [31] suggest that students' attention and concentration seem to increase during PAL lessons.

These findings are consistent with existing research on experiential learning methods, which emphasizes the benefits of active participation in the learning process. For example, Uyen et al. [40] found that sixth grade students engaged in experiential learning in mathematics showed significant improvements in both academic achievement and enthusiasm. Similarly, the review by Burch et al. [41] supports these findings by providing conclusive results on the impact of experiential learning on improving academic outcomes. Another example is project-based learning, which has been shown to have a positive impact on academic performance and content retention in both primary [42] and secondary education [43]. Finally, Okpala & Okigbo [44] found that methods such as role-playing significantly increased student achievement in secondary education. These similarities suggest that PAL, like other experiential approaches, promotes deeper understanding and better knowledge retention, reinforcing the notion that a more interactive and engaging learning environment can significantly enhance students' learning experiences without compromising content.

The shift towards cooperative activities within PAL environments has proved to be crucial in promoting cooperation between students and improving interpersonal relationships. This is reflected in comments from teachers and students who noted an improvement in socialization and cooperation during PAL lessons. In addition, they noted that the cooperative nature of PAL activities led to increased participation and provided opportunities for conflict resolution, which is less common in traditional classrooms due to less interaction between students. These findings are consistent with previous experiences reported by teachers in primary [45] and secondary [31] education. Both studies highlighted an improvement in students' collaboration, recognizing the group work inherent in PAL as a facilitator of peer learning and knowledge sharing, which encourages greater participation. In this respect, Lerum et al. [31] also found in their research that more students took the risk of participating in PAL tasks compared to traditional teaching.

The improvement observed in students' relationships may also be related to the delivery of outdoor education. Focusing on this context, research by Fägerstam et al. [46] evaluated secondary school teachers' experiences of outdoor learning environments, highlighting not only the academic improvements resulting from more collaborative learning, but also the potential for deepening relationships between teachers and students. These findings highlight the social and collaborative benefits of PAL and position it as a valuable pedagogical approach for improving student interactions and classroom dynamics.

An important aspect of the sustainability of the intervention over time is teachers' perceptions of its feasibility. In this sense, the results of our research show that teachers' responses revolve around three key aspects: resources, frequency and timing.

Firstly, teachers expressed confidence in their ability to develop PAL lessons autonomously, largely due to the initial support and resources provided. However, previous experiences of teachers participating in PAL interventions highlight that the lack of adequate training and additional resources can be a significant barrier to the continued adoption of PAL [28]. At time, teachers even feel overwhelmed by the complexity of integrating physical activity into their lessons while ensuring learning outcomes [30]. This discrepancy with the experiences shown in the present study seems to be due to the fact that the ongoing support from the research team was crucial in increasing this confidence, highlighting the need to provide teachers with sufficient resources throughout the intervention. Secondly, teachers considered PAL to be a valuable pedagogical tool, but more suitable for occasional rather than daily use. This view is supported by the testimonies provided in the research by Schmidt et al. [30], where teachers understand PAL as a tool to be used occasionally when students need variation. Finally, teachers suggested that the PAL could be particularly effective in specific moments, such

as the introduction of new content or review sessions. Although the current literature has not extensively investigated the optimal time to implement PAL, employing it in situations that complement and reinforce learning, and aligning it with students' needs, is critical to maximizing its impact and effectiveness.

To date, the implementation of PAL in secondary education continues to present significant challenges. Addressing these challenges by prioritizing the critical issues discussed in our study, such as adequate resource allocation, teacher training and the adoption of flexible inclusion strategies, could facilitate the successful implementation of PAL in more schools in the future. Incorporating these considerations would not only improve the sustainability and effectiveness of PAL but also allow students to benefit from the positive educational and social impacts that its implementation appears to generate. These include improvements in students' motivation, perceptions of more meaningful learning in the PAL lessons, and increased cooperation and socialization among students.

## Limitations and strengths

The present study is not without some limitations. Firstly, for logistical reasons, the focus groups were conducted with a randomized sub-sample of students. This may affect the representativeness of the results, as the inclusion of the total sample could have provided a completer and more diverse picture of their perceptions and experiences. Although the authors feel that they have reached saturation in terms of data generation, the results should not be generalized to all Spanish schools. Nevertheless, the consistency of themes across both participating schools suggests potential transferability. Another limitation is the support provided by the research team during the intervention, which, although valuable in ensuring fidelity, may have influenced teachers' experiences by facilitating the implementation of the approach, as well as limiting its scalability. Implementation of PAL on a larger scale would require investment in teacher training and provision of resources. Finally, the study highlights the challenge of addressing different learning paces in PAL classes. Students who struggle with the subject may find it more difficult to engage, which could impact both individual learning and group dynamics. To ensure inclusive implementation, as well as the viability and sustainability of PAL, adaptable and sufficient teacher support strategies and resources are needed.

However, the study also has several significant strengths that add value to the research. Firstly, it is the first study in secondary education to assess both teachers' and students' perceptions of PAL, providing a valuable contrast of perspectives and a more complete picture of the implementation and impact of the intervention. In addition, the sample included a heterogeneous range of teachers with different profiles and ages, as well as students randomly selected from different classes, with an equal gender distribution. This selection process not only enriched the diversity of perspectives on PAL lessons but also facilitated the inclusion of students with different levels of engagement, allowing for a fuller understanding of different views and experiences. None of the participants had previous experience of PAL, which provided a fresh and unbiased perspective. Finally, the interviews were conducted by the same people in each school who had been present throughout the intervention. This not only made participants feel more comfortable in expressing their experiences, but also ensured consistency in data collection, thereby increasing the reliability of the findings.

## Conclusion

This study provides a comprehensive overview of the impact of PAL in secondary education, examining its effects on student motivation, quality of learning and social interaction. Based on the perceptions and experiences of teachers and students, the findings suggest that this method can not only improve student motivation but can also have a positive impact on the engagement and behavior of some students. Furthermore, although PAL lessons appear to cover less curriculum content than traditional lessons, they promote significantly deeper understanding and a higher quality of learning, suggesting that although the scope of content is narrower, the depth of understanding achieved is significantly greater. Both teachers and pupils emphasized that PAL lessons promoted co-operation and socialization between students, underlining their ability to promote meaningful peer interaction in the educational context.

These findings highlight the importance of involving the educational community in the adoption of PAL and integrating this pedagogical approach into a comprehensive educational framework to maximize its effectiveness and long-term sustainability. Collectively, the findings underscore key areas for future research and policy development aimed at incorporating active methods to optimize student learning and engagement in secondary education.

## Acknowledgments

The authors gratefully acknowledge the participation and collaboration of all schoolchildren, teachers, school leaders and families of the two schools involved in the present study.

## Author contributions

**Conceptualization:** Alberto Grao-Cruces, Daniel Camiletti-Moirón, David Sánchez-Oliva.

**Data curation:** María González-Pérez, Francisco J. Bandera-Campos, Anna Chalkley.

**Formal analysis:** María González-Pérez, David Sánchez-Oliva.

**Funding acquisition:** Alberto Grao-Cruces, Daniel Camiletti-Moirón, David Sánchez-Oliva.

**Investigation:** María González-Pérez, Alberto Grao-Cruces, Francisco J. Bandera-Campos, Daniel Camiletti-Moirón, David Sánchez-Oliva.

**Methodology:** Alberto Grao-Cruces, Daniel Camiletti-Moirón, David Sánchez-Oliva.

**Project administration:** Alberto Grao-Cruces, Daniel Camiletti-Moirón, David Sánchez-Oliva.

**Supervision:** Alberto Grao-Cruces, Daniel Camiletti-Moirón, David Sánchez-Oliva.

**Visualization:** María González-Pérez, Alberto Grao-Cruces, David Sánchez-Oliva.

**Writing – original draft:** María González-Pérez.

**Writing – review & editing:** María González-Pérez, Alberto Grao-Cruces, Francisco J. Bandera-Campos, Anna Chalkley, Daniel Camiletti-Moirón, David Sánchez-Oliva.

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
