## [Decision Letter · Decision Letter 0]

28 Jul 2025

Dear Dr. Grao-Cruces,

Thank you for submitting your manuscript to PLOS ONE. After careful consideration, we feel that it has merit but does not fully meet PLOS ONE’s publication criteria as it currently stands. Therefore, we invite you to submit a revised version of the manuscript that addresses the points raised during the review process.

We look forward to receiving your revised manuscript.

Kind regards,

Francesca D'Elia, Ph.D.

Academic Editor

PLOS ONE

Journal Requirements:

2. Thank you for stating the following financial disclosure: [This work was supported by The National Plan for Research, Development, and Innovation (R&D&I) from the Spanish Ministry of Science, Innovation and Universities [PID2019-104023RA-I00] (DCM); The Andalusian Plan for R&D&I Regional Ministry of Economy, Knowledge, Enterprise and University of Andalusia [P20_00908] (AGC); and the Junta de Extremadura & Fondos Feder [IB20126] (DSO). MGP and FJBC are supported by a grant from the Spanish Ministry of Education [FPU22/01430 and FPU21/03385 respectively].]. 

Additional Editor Comments:

This qualitative study explores the perceptions of teachers and students following a 16-week intervention using Physically Active Learning (PAL) in secondary math education. Findings indicate PAL increases student motivation, encourages deeper understanding and improves social interaction. Teachers see PAL as feasible, especially with proper support, but recommend using it selectively rather than as a fixed methodology.

Suggested Minor Revisions

• Affiliation: ensure consistency in listing affiliations (e.g., some institutions are listed with full addresses, others are abbreviated).

• Abstract: remove redundancy (e.g. “better content retention and understanding” could be simplified to “enhanced retention”).

• Method Section:

Table 1 Caption Revision Suggestion. The current caption for Table 1 (“Summary of the sample of participants”) does not accurately reflect the data presented. Since the table specifically details the sub-sample of students who participated in the focus groups, as well as the teachers and researchers involved, it is recommended to revise the caption for clarity. Suggested alternative: “Participants in the focus groups: sub-sample overview.”

Data analysis. Clarify whether data saturation was planned or emerged organically.

• Results Section: replace phrases like “PAL helped some students to keep up with the pace of the class” with more precise language such as “PAL supported student pacing for those who typically struggle with classroom instruction.”

• Discussion: strengthen transitions between studies cited, some paragraphs begin abruptly with literature comparisons, and clarify the role novelty played in the motivation observed, currently mentioned but not critically examined.

• Conclusion: rephrase “Taken together, the findings…” to something sharper like “Collectively, the findings underscore…”

• Formatting & Language:

- Fix small typos: e.g., “seasonable” in Research data availability should likely be “reasonable.”

- Ensure consistent formatting in references: use either “Available at” or DOI, not both inconsistently.

Reviewers' comments:

Reviewer's Responses to Questions

**Comments to the Author**

1. Is the manuscript technically sound, and do the data support the conclusions?

Reviewer #1: Yes

Reviewer #2: Yes

2. Has the statistical analysis been performed appropriately and rigorously?

Reviewer #1: Yes

Reviewer #2: Yes

3. Have the authors made all data underlying the findings in their manuscript fully available?

Reviewer #1: Yes

Reviewer #2: Yes

4. Is the manuscript presented in an intelligible fashion and written in standard English?

Reviewer #1: Yes

Reviewer #2: Yes

Reviewer #1: I consider it to be a rigorous and high-quality work. It is true that some of the articles published by you have very similar results to those presented in this paper (Example: A mixed-methods approach of the effect of physically active learning on time-on-1 task in the secondary education class: the ACTIVE CLASS study). Understanding that an exhaustive bibliographic review has been carried out and is thus contemplated in the references section, however there is a study from Finland very similar to yours that it would be advisable to include:

Sirpa Sneck, Heidi Syväoja, Sanna Järvelä & Tuija Tammelin (2023) More active lessons: teachers’ perceptions of student engagement during physically active maths lessons in Finland, Education Inquiry, 14:4, 458-479, DOI:10.1080/20004508.2022.2058166

In another order of analysis of your article, you have been able to observe that in line 115, they set up a chair saying: "In addition, no study to date included perspectives of students and teachers together" which is not entirely true, considering that in the journal PLOS ONE, there is an article with that double approach that you refer to as an essential value of your studies. I therefore recommend that you review this article and take into account the reformulation of some of its sentences.

Simard L, Bouchard J, Lavallière M, Chevrette T. Promoting physical activity and academic achievement through physically active learning: Qualitative perspectives of co-design and implementation processes. PLoS One. 2023 Nov 22;18(11):e0294422. doi: 10.1371/journal.pone.0294422. PMID: 37992080; PMCID: PMC10664963

Reviewer #2: The manuscript addresses a highly relevant and important topic. The article explores Physically Active Learning (PAL), a pedagogical approach that integrates physical activity into core subject lessons to combine physical and academic benefits. Despite its potential, implementing PAL in secondary education remains challenging, making it crucial to understand teachers' and students' perceptions and experiences. The study's primary aim was to assess the perceptions and experiences of teachers and students after participating in a PAL intervention in secondary education. From the point of view of methodological rigor, the problem of the study was identified, referring its objectives, and presented scientific evidence on the subject of the study. The research variables are defined and the procedures and instruments for the research should more well explained. The results are presented, being complemented with a discussion. The conclusions are very enlightening and the references are adjusted to the topic under discussion.

Few suggestions to improve the manuscript:

“Setting and participants” line 148: the number of participants in the study are not clear. The authors written “ …A total of 104 students 148 (aged 12-14 years, n= 45 boys) and five mathematics teachers (two male) were included in the study…” what happened to the others 103 students? And were all from the same gender?

“Setting and participants” line 159, table 1: why is important to include the information about the two researchers? Why there is a frequency of each student? Why the authors don´t present the mean and SD of each variable concerning participants (i.e..: Age, sex. School year)? Concerning teachers the authors should explain better the meaning of a teacher belong a some class, what means class 1, class 4 and 5…

Should be explain some data about the control group, how many students, age, sex, school year…

It´s important to explain how many schools were involved in the program.

“Intervention” lines 164, 165 – the PAL program should be more well explained. For example: How many times per week and minutes was the program? Concerning the physical conditions variables, where they all integrate in the program. The experimental group how many traditional (in room) math class have and how many have in outdoor space per week? And the control group?

“Data Collection” lines 204, 205 – the question of the interview is repeated.

The development and validation process of the instruments-interviews (students and teachers) should be more well explained.

Could be interesting to understand if the academic results were similar between the students that integrate the experimental and control groups.

The chapter “Limitations and strengths” it is very long it is suggested to be shorter.

The manuscript should respect the structure defined by Plos One guidelines.

**Do you want your identity to be public for this peer review?** For information about this choice, including consent withdrawal, please see our Privacy Policy

Reviewer #1: No

Reviewer #2: No

---

## [Author Response · Author response to Decision Letter 1]

1 Oct 2025

Dear Editor Ph.D. Francesca D’Elia,

Thank you for reviewing our manuscript and for providing us with the reviewers' comments. We are very grateful for the time and effort you and the other reviewers have devoted to evaluating our work.

We understand and recognize the importance of the concerns raised in improving the quality of the manuscript. We have carefully considered all the comments and hope that the revised version addresses each of your concerns thoroughly. We are optimistic that these revisions will strengthen the manuscript and align it with the standards of Plos One.

You will find a point-by-point response to the comments below, starting with yours and continuing with those of each reviewer. All changes to the original manuscript have been highlighted using the track change’s function.

Thank you once again for the opportunity to submit a revised version of the manuscript.

Yours sincerely,

EDITOR

Comment:

Affiliation: ensure consistency in listing affiliations (e.g., some institutions are listed with full addresses, others are abbreviated)

Answer:

We have reviewed the list of affiliations and modified it to ensure consistency in the use of abbreviations and addresses. We would like to clarify that the name “GALENO” is not an abbreviation but the full name. On the other hand, the language is established according to the internal rules of each institution (lines 8-18):

“1GALENO Research Group, Department of Physical Education, Faculty of Education Sciences, University of Cadiz, Puerto Real, Spain.

2 Instituto de Investigación e Innovación Biomédica de Cádiz (INiBICA), Cadiz, Spain.

3Faculty of Health Studies, University of Bradford, Richmond Road, Bradford, United Kingdom.

4Centre for Applied Education Research, Wolfson Centre for Applied Health Research, Bradford Teaching Hospitals National Health Service Foundation Trust, Bradford Royal Infirmary, Duckworth Lane, Bradford, United Kingdom.

5 Análisis Comportamental de la Actividad Física y el Deporte (ACAFYDE) research group, Department of Didactics of Musical, Plastic and Body Expression, Faculty of Sports Sciences, University of Extremadura, Cáceres, Spain”.

Comment:

Abstract: remove redundancy (e.g. “better content retention and understanding” could be simplified to “enhanced retention”).

Answer:

Thank you very much for your feedback. We have removed the redundancy by replacing the case in question with the term suggested by the editor (lines 37–41):

“Although teachers and students perceived that the amount of content covered in a PAL lesson was lower compared to traditional lessons, they emphasized that learning was more meaningful due to enhanced retention, facilitated by active engagement and dynamic teaching approaches”.

Comment:

Method Section: Table 1 Caption Revision Suggestion. The current caption for Table 1 (“Summary of the sample of participants”) does not accurately reflect the data presented. Since the table specifically details the sub-sample of students who participated in the focus groups, as well as the teachers and researchers involved, it is recommended to revise the caption for clarity. Suggested alternative: “Participants in the focus groups: sub-sample overview.”

Answer:

Thank you for your comment. We agree that the original title did not accurately reflect the content of the table. We have therefore updated the title to: “Participants in the interviews and focus groups: sub-sample overview”. We have chosen to include the word “interviews” in the suggested proposal to adequately represent the participation of both students and teachers, since, as the editor mentions, the sample of teachers participating in the interviews is also reflected in the table.

Comment:

Data analysis. Clarify whether data saturation was planned or emerged organically.

Answer:

Thank you for your comment. We appreciate the opportunity to clarify this point. Our intention with this study was not to achieve saturation but rather to interview all the participants who consented to be interviewed to achieve information power [1]. Typically, participants are recruited until the data reach a level of saturation (i.e., wherein no new themes are identified through further data collection [2]. However, saturation is affected by several factors including the quality of the interviews and the scope of the study. The number of interviews and focus groups was selected in advance based on the study’s scope and the research team’s logistical capacity. However, as the analysis progressed, we observed that no new themes emerged after the third teacher interview and the fifth focus group, indicating that thematic saturation had been reached. Despite this, all six focus groups and five interviews were analyzed in full to ensure the richness and diversity of participants’ perspectives was fully captured. We have clarified this information in lines (250-256) of the manuscript:

“Although the number of interviews and focus groups was determined in advance based on the study’s scope and available resources, data saturation was identified organically during the analysis, when no new themes or categories emerged. Saturation was perceived to have been achieved after the 5th focus group and the 3rd interview, where recurring patterns and consistent themes were observed across both teachers' and students' responses. This repetition of the findings provided confidence that the data adequately captured the range of experiences and perceptions of the participants”.

1. Malterud K, Siersma VD, Guassora AD. Sample Size in Qualitative Interview Studies: Guided by Information Power. Qualitative Health Research. 2015;26(13):1753-1760. doi.org/10.1177/1049732315617

2. Fusch L & Ness, P. Are we there yet? Data saturation in qualitative research. Qualitative Report. 2015; 20(9), 1409-1416.

Comment:

Results Section: replace phrases like “PAL helped some students to keep up with the pace of the class” with more precise language such as “PAL supported student pacing for those who typically struggle with classroom instruction.”

Answer:

Thank you for your comment. We have modified this information in accordance with your suggestion.

Comment:

Discussion: strengthen transitions between studies cited, some paragraphs begin abruptly with literature comparisons and clarify the role novelty played in the motivation observed, currently mentioned but not critically examined.

Answer:

Thank you very much for your observations. We have reviewed the discussion section and made changes across it to ensure greater consistency and coherence compared with the existing literature.

Additionally, we have expanded the information regarding the potential role of novelty in the observed motivation, as suggested by the editor (lines 438–444):

“However, it is important to consider whether the motivational benefits of PAL may be partially linked to its novel approach, as the satisfaction of the need for novelty seems to predict intrinsic motivation, as well as students' participation and intention to engage in physical activity [37,38]. As in physical education, this suggests that introducing new and varied proposals, such as PALs, can increase motivation precisely because they satisfy this basic psychological need, and therefore, if overused, its impact on engagement and learning could diminish over time”.

Comment:

Conclusion: rephrase “Taken together, the findings…” to something sharper like “Collectively, the findings underscore…”

Answer:

Thank you for your feedback. We have rephrased this statement in accordance with your suggestion.

Comments:

Formatting & Language:

- Fix small typos: e.g., “seasonable” in Research data availability should likely be “reasonable.”

- Ensure consistent formatting in references: use either “Available at” or DOI, not both inconsistently.

Answer:

Thank you very much for your comments. We have corrected the error identified by the editor and thoroughly reviewed and amended the list of references to ensure consistent formatting.

REVIEWER 1

Comment:

I consider it to be a rigorous and high-quality work. It is true that some of the articles published by you have very similar results to those presented in this paper (Example: A mixed-methods approach of the effect of physically active learning on time-on-1 task in the secondary education class: the ACTIVE CLASS study). Understanding that an exhaustive bibliographic review has been carried out and is thus contemplated in the references section, however there is a study from Finland very similar to yours that it would be advisable to include:

Sirpa Sneck, Heidi Syväoja, Sanna Järvelä & Tuija Tammelin (2023) More active lessons: teachers’ perceptions of student engagement during physically active maths lessons in Finland, Education Inquiry, 14:4, 458-479, DOI:10.1080/20004508.2022.2058166

Answer:

Thank you very much for your kind comments and for taking the time to review our manuscript. As you point out, our previously published article “A mixed-methods approach to the effect of physically active learning on time-on-task in secondary education classes: the ACTIVE CLASS study”, presents qualitative results derived from teacher interviews. However, we believe that, while there are similarities in areas such as student behavior and participation, the present study makes a substantially different contribution. Specifically, it provides a broader and deeper insight into the implementation of PAL by integrating the perceptions of both teachers and participating students. Unlike the aforementioned study, which exclusively extracted teachers' perceptions related to time-on-task, this study covers additional aspects such as motivation, depth of learning, social interaction and the feasibility of the intervention. As the reviewer understands, this study is cited in the references section.

Regarding the study by Sneck et al. (2023) that was suggested by the reviewer, as the reviewer himself points out, it is an article similar to the present research, with the peculiarities that it analyses only the perspectives of teachers and not those of students, and that it is developed in primary education, unlike our research, which is developed in secondary education. This study was cited in our manuscript as reference 25. However, we have reviewed the information in our manuscript regarding this study and expanded upon it in lines (93–102):

“This qualitative perspective has been addressed by several studies in primary education, which focus separately on teachers' perceptions [25,26], students' perceptions [27] and some that combine both perspectives [28,29]. The main findings of these studies indicate an increase in students' engagement and concentration during PAL lessons [25]. They also consider PAL as a method that introduces variation and enjoyment into learning [25,28,29], improves social relationships between peers and promoting quality learning [25,26,29]. However, some teachers have difficulty identifying its cognitive and social benefits when applying it to students with cognitive challenges due to their support needs. Furthermore, although they report that it is another path to learning, they are uncertain about achieving learning objectives [25]”.

Comment:

In another order of analysis of your article, you have been able to observe that in line 115, they set up a chair saying: "In addition, no study to date included perspectives of students and teachers together" which is not entirely true, considering that in the journal PLOS ONE, there is an article with that double approach that you refer to as an essential value of your studies. I therefore recommend that you review this article and take into account the reformulation of some of its sentences.

Simard L, Bouchard J, Lavallière M, Chevrette T. Promoting physical activity and academic achievement through physically active learning: Qualitative perspectives of co-design and implementation processes. PLoS One. 2023 Nov 22;18(11):e0294422. doi: 10.1371/journal.pone.0294422. PMID: 37992080; PMCID: PMC10664963

Answer:

Thank you for your comment. As the reviewer rightly points out, there are already studies that evaluate the perceptions of both teachers and students of PAL. However, these studies have focused on primary education. In the highlighted sentence, we intended to refer to the context of secondary education, as, to our knowledge, no studies have yet been conducted that include both perspectives in this context. We understand that this sentence may be confusing, so we have made minor changes to the sentence to clarify this (lines 116–118):

“Despite the emerging research on PAL in secondary education, an important gap in understanding its impact and implementation remains. Furthermore, no study to date has included both students and teacher perspectives together in secondary education”.

REVIEWER 2

Comment:

The manuscript addresses a highly relevant and important topic. The article explores Physically Active Learning (PAL), a pedagogical approach that integrates physical activity into core subject lessons to combine physical and academic benefits. Despite its potential, implementing PAL in secondary education remains challenging, making it crucial to understand teachers' and students' perceptions and experiences. The study's primary aim was to assess the perceptions and experiences of teachers and students after participating in a PAL intervention in secondary education. From the point of view of methodological rigor, the problem of the study was identified, referring its objectives, and presented scientific evidence on the subject of the study. The research variables are defined and the procedures and instruments for the research should more well explained. The results are presented, being complemented with a discussion. The conclusions are very enlightening and the references are adjusted to the topic under discussion.

Answer:

Thank you very much for your comments and for the time and effort you have put into reviewing our manuscript. We have considered and reviewed each of your suggestions. Below, we respond to each of them individually.

Comment:

“Setting and participants” line 148: the number of participants in the study are not clear. The authors written “ …A total of 104 students 148 (aged 12-14 years, n= 45 boys) and five mathematics teachers (two male) were included in the study…” what happened to the others 103 students? And were all from the same gender?

Answer:

Thank you for your feedback. We would like to clarify that, as mentioned in the design section, this particular study emerged from the ACTIVE CLASS project, in which the group that implemented the physically active learning intervention consisted of a total of 104 students. Of those 104 participants, this qualitative study, due to its nature, included a subsample of 36 students (n= 18 boys), as it could not cover the entire sample of the intervention group. In order to avoid confusion, we have slightly modified this information in lines (149-159):

“Participant recruitment took place from November 2022 to January 2023. The study was explained to the participants before starting, and the volunteers, parents, or tutors provided written informed consent. A total of 104 students (aged 12-14 years, n= 45 boys) and five mathematics teachers (two male) participated in the PAL intervention. To be included in the analysis, all teachers involved in the intervention were invited to participate in the interviews. On the other hand, from the total number of students, a sub-sample of 36 (six students per participating class [n= 18 boys]) were invited to participate and completed the focus groups for data collection (table 1). This sub-sample of students was randomly selected within each participating class, with a balanced gender distribution to ensure homogeneity in the composition of the group. Neither students nor teachers had been exposed to previous PAL experiences”.

Additionally, we have modified the title of Table 1 to: "Participants in the interviews and focus group: sub-sample overview”.

Comment:

“Setting and participants” line 159, table 1: why is important to include the information about the two researchers? Why there is a frequency of each student? Why the authors don´t pr

---

## [Editor Report · Decision Letter 1]

29 Oct 2025

Less is more: qualitative insights into physically active learning in secondary math education

PONE-D-25-29708R1

Dear Dr. Grao-Cruces,

We’re pleased to inform you that your manuscript has been judged scientifically suitable for publication and will be formally accepted for publication once it meets all outstanding technical requirements.

Kind regards,

Francesca D'Elia, Ph.D.

Academic Editor

PLOS ONE

Additional Editor Comments (optional):

Dear Authors,

thank you for submitting the revised version of your manuscript entitled “Less is More: Qualitative Insights into Physically Active Learning in Secondary Math Education.”

We appreciate the care with which you addressed the reviewers’ feedback. The revisions have clearly strengthened the manuscript, enhancing its clarity, methodological transparency, and alignment with PLOS ONE’s editorial standards. Your refinements to the abstract, results, and discussion sections, along with the improved explanation of coding and saturation, contribute to a more robust and readable presentation of your findings.

We also acknowledge the thoughtful clarifications regarding sample composition, the PAL intervention structure, and the development of interview tools. These additions help contextualize the study and reinforce its relevance.

With these improvements in place, we are pleased to accept your manuscript for publication. Congratulations, and thank you for your contribution to the field.

Kind regards,

the Academic Editor
---

## [Editor Report · Acceptance letter]

PONE-D-25-29708R1

PLOS ONE

Dear Dr. Grao-Cruces,

I'm pleased to inform you that your manuscript has been deemed suitable for publication in PLOS ONE. Congratulations! Your manuscript is now being handed over to our production team.

Kind regards,

on behalf of

Dr. Francesca D'Elia

Academic Editor

PLOS ONE